# Association between Thyroid Cancer and Breast Cancer: Two Longitudinal Follow-Up Studies Using a National Health Screening Cohort

**DOI:** 10.3390/jpm12020133

**Published:** 2022-01-20

**Authors:** Young Ju Jin, Mi Jung Kwon, Ji Hee Kim, Joo-Hee Kim, Hyo Geun Choi

**Affiliations:** 1Department of Otorhinolaryngology-Head & Neck Surgery, Wonkwang University Hospital, College of Medicine, Wonkwang University, Iksan 54538, Korea; chindol1@wku.ac.kr; 2Department of Pathology, Hallym University Sacred Heart Hospital, College of Medicine, Hallym University, Anyang 14068, Korea; mulank99@hallym.or.kr; 3Department of Neurosurgery, Hallym University Sacred Heart Hospital, College of Medicine, Hallym University, Anyang 14068, Korea; kimjihee@hallym.or.kr; 4Division of Pulmonary, Allergy and Critical Care Medicine, Department of Medicine, Hallym University Sacred Heart Hospital, College of Medicine, Hallym University, Anyang 14068, Korea; luxjhee@hallym.or.kr; 5Department of Otorhinolaryngology-Head & Neck Surgery, Hallym University Sacred Heart Hospital, College of Medicine, Hallym University, Anyang 14068, Korea

**Keywords:** thyroid cancer, breast cancer, radioactive iodine treatment, longitudinal follow-up study, national health screening cohort

## Abstract

**Background**: The purpose of this study was to evaluate the association between thyroid cancer and breast cancer. **Methods**: Data from the Korean National Health Insurance Service-Health Screening Cohort were collected from 2002 to 2013. In study I, 3949 thyroid cancer participants were 1:4 matched with 15,796 control I participants, and hazard ratios (HRs) with 95% confidence intervals (CIs) for breast cancer were evaluated using a stratified Cox proportional hazard model. In study II, 3308 breast cancer participants were 1:4 matched with 13,232 control II participants, and HRs with 95% CIs for thyroid cancer were assessed in the same way as in study I. In the subgroup analyses, associations were analyzed according to radioactive iodine (RAI) treatment and age (<60 years old and ≥60 years old). **Results**: The adjusted HR for breast cancer in the thyroid cancer group was 1.64 (95% CI = 1.13–2.39, *p* = 0.010). The adjusted HR for thyroid cancer in the breast cancer group was 1.91 (95% CI = 1.47–2.49, *p* < 0.001). In the subgroup analyses, the groups that were older and not treated with RAI treatment showed consistent results in study I, and the younger and older groups showed consistent results in study II. **Conclusions**: Based on this cohort study, breast and thyroid cancer have a reciprocal positive association.

## 1. Introduction

Breast cancer and thyroid cancer are common malignancies in females. Breast cancer is the most common cancer, accounting for 30% of new cancer cases in females, followed by lung cancer (12%), colon cancer (8%), uterine corpus cancer (7%), and thyroid cancer (4%) in the US [1]. In Korea, the incidence by cancer site was highest in breast cancer (20.3%), thyroid cancer (18.3%), colon cancer (10.4%), stomach cancer (8.9%), and lung cancer (7.63%) in that order [2].

The annual breast cancer incidence rate varied from 84.7 cases per 100,000 females to 127.3 cases per 100,000 females in Asian Americans/Pacific Islanders and non-Hispanic white women, respectively, from 2006 to 2010 [3]. The incidence of thyroid cancer in the US increased by 211% from 1975 to 2013 [4], and in Korea, it increased from 7.2 cases per 100,000 population to 51.0 per 100,000 population from 1994 to 2016 [2,5]. Both breast cancer and thyroid cancer patients showed increased 5-year survival rates at 93.2% and 100.1%, respectively, from 2013–2017 due to advancements in cancer treatment and early detection [2]. 

Because breast and thyroid cancer patients generally have a long life expectancy, recognizing risk factors for second primary malignancy is very important. Several epidemiological studies have suggested a positive link between thyroid cancer and breast cancer [6,7]. Some studies have suggested an increased risk of thyroid cancer in patients with a history of breast cancer [8,9], while other studies have reported an increased risk of breast cancer in patients with a history of thyroid cancer [10,11]. However, the specific mechanisms responsible for the association between thyroid cancer and breast cancer have not been clearly identified. Possible causes have been suggested, including genetic, environmental, and treatment-related factors, such as radioactive iodine (RAI) treatment and radiotherapy, tumor suppression gene-related factors, smoking or drinking habits, and obesity-related factors [12]. 

Many studies have been performed to identify possible common risk factors for thyroid cancer and breast cancer. Cigarette smoking and alcohol consumption are very common social habits that expose people to known carcinogens that cause many kinds of cancer. Some studies have reported that these two modifiable social habits influence the risk of thyroid cancer and breast cancer by altering thyroid-stimulating hormone (TSH) levels and sex steroid hormone levels, producing direct carcinogenic effects on mammary gland tissue and influencing body weight change [12,13]. Moreover, the number of obese individuals has increased in recent decades, and obesity has been reported to be positively associated with increased risk for both thyroid and breast cancer [14,15]. 

We hypothesized that there is an association between thyroid cancer and breast cancer, and this association is caused by common etiologic factors. The aim of this study was to evaluate the association between thyroid cancer and breast cancer in a nationwide population-based cohort study using data from the Korean National Health Insurance Service-Health Screening Cohort (NHIS-HEALS). The hazard ratios (HRs) for breast cancer in the thyroid cancer group and thyroid cancer in the breast cancer group were evaluated to identify the reciprocal relations between two diseases. The Charlson Comorbidity Index (CCI) score, obesity status, smoking status, and alcohol consumption status were adjusted as covariates. In the subgroup analysis, the HRs for breast cancer in the thyroid cancer group and thyroid cancer in the breast cancer group were evaluated according to radioactive iodine (RAI) treatment and age.

## 2. Materials and Methods

### 2.1. Study Population

The ethics committee of Hallym University (2017-I102) permitted this study. Written informed consent was waived by the Institutional Review Board. All analyses adhered to the guidelines and regulations of the ethics committee of Hallym University. A detailed description of the Korean National Health Insurance Service-Health Screening Cohort data is described elsewhere [16]. 

### 2.2. Definition of Thyroid Cancer

Thyroid cancer was defined by the International Classification of Diseases 10th revision (ICD-10) code C73 in the medical records and treatment by thyroidectomy. Thyroidectomy was defined by operation codes P4551 and P4553 (hemithyroidectomy) and P4442, P4554, and P4561 (total thyroidectomy). Thyroid cancer patients were classified according to RAI treatment (treatment code: HD071) following the classification method in our previous study [17]. 

### 2.3. Definition of Breast Cancer

Breast cancer was defined as ≥2 ICD-10 C50 codes in the medical record or death due to breast cancer following the method in our previous study [18].

### 2.4. Participant Selection

#### 2.4.1. Study I

Thyroid cancer patients were selected from 514,866 participants with 497,931,549 medical claim codes (*n* = 5075). The control group comprised participants who were never diagnosed with having thyroid cancer from 2002 to 2013 (*n* = 509,791). Since most of the breast cancer patients were women, men were excluded from both the thyroid cancer group (*n* = 1044) and the control group (*n* = 278,081). Thyroid cancer patients were 1:4 matched with control participants for age, income, and region of residence. To minimize selection bias, the control participants were selected with a random number method. The index date of each thyroid cancer patient was set as the time of treatment of thyroid cancer. The index dates of the control participants were set as the index dates of their matched thyroid cancer patients. Therefore, each matched thyroid cancer patient with control participants had the same index date. Thyroid cancer patients with breast cancer before the index date were excluded (*n* = 82). Control participants with breast cancer before the index date were also excluded. Among the control participants, 215,914 were excluded during the matching procedure. Finally, 3949 patients in the thyroid cancer group were 1:4 matched with 15,796 patients in the control I group (Figure 1).

#### 2.4.2. Study II

Breast cancer patients were selected from 514,866 participants with 497,931,549 medical claim codes (*n* = 3376). The control group comprised participants who were never identified as having breast cancer from 2002 to 2013 (*n* = 511,490). Since most of the breast cancer patients were women, men were excluded from both the breast cancer group (*n* = 31) and the control group (*n* = 279,094). Breast cancer patients were 1:4 matched with control participants for age, income, and region of residence. To minimize selection bias, the control participants were selected with a new random number procedure. The index date of each breast cancer patient was set as the time of treatment of breast cancer. The index dates of the control participants were set as the index dates of their matched breast cancer patients. Therefore, each breast cancer patient matched with control participants had the same index date. Breast cancer patients with thyroid cancer before the index date were excluded (*n* = 37). Control participants with thyroid cancer before the index date were also excluded. Among the control participants, 219,164 were excluded during the matching procedure. Finally, 3308 breast cancer patients were 1:4 matched with 13,232 patients in the control II group (Figure 2).

### 2.5. Covariates

Age groups were divided into 5-year intervals: 40–44, 45–49, 50–54…, and 85+ years old (total of 10 age groups). Income groups were classified into 5 classes (from class 1 (lowest income) to class 5 (highest income)). The region of residence was grouped into urban and rural areas, following the methods in our previous study [18].

Tobacco smoking, alcohol consumption, and obesity identified by body mass index (BMI, kg/m^2^) were categorized in the same ways as those in our previous study [19]. The CCI has been used widely to measure disease burden, considering 17 comorbidities as continuous variables (total score from 0 (no comorbidities) to 29 (multiple comorbidities)) [20]. In our study, the scores were calculated without thyroid cancer and breast cancer (Appendix A).

### 2.6. Statistical Analyses

The general characteristics between the thyroid cancer and control I groups in study I and the breast cancer and control II groups in study II were compared using chi-square tests.

To analyze the HRs with CIs, a stratified Cox proportional hazard model was used to compare the risk of breast cancer in the thyroid cancer group with that in the control I group (study I) and to compare the risk of thyroid cancer in the breast cancer group with that in the control II group (study II). In these analyses, the crude model and the model adjusted for obesity, smoking status, alcohol consumption, and CCI score were calculated. The analyses were stratified for matching variables, including age, income, and region of residence.

For the subgroup analyses, we divided participants by age (<60 years old and ≥60 years old) in study I and study II. In study I, we divided participants according to RAI treatment (with RAI and without RAI) for subgroup analysis (Figure 1). The crude and adjusted models were analyzed in the subgroup analyses.

The Kaplan-Meier method and the log-rank test were used to analyze the cumulative incidence rates of breast cancer in thyroid cancer participants and the control I group (Figure 3) and the cumulative incidence rates of thyroid cancer in breast cancer participants and the control II group (Figure 4).

Two-tailed analyses were performed, and significance was defined as *p* values less than 0.05. SAS version 9.4 (SAS Institute Inc., Cary, NC, USA) was used for the statistical analyses.

## 3. Results

### 3.1. Detailed Descriptions

#### 3.1.1. Study I. Risk of Breast Cancer in the Thyroid Cancer Group

Age, income, and region of residence were exactly matched between the thyroid cancer and control I groups (*p* = 1.000). The rates of high CCI scores, overweight, obesity I, obesity II, nonsmoking status, and alcohol consumption < 1 time a week were higher in the thyroid cancer group than in the control I group (each *p* < 0.05). The ratio of breast cancer was higher in the thyroid cancer group than in the control I group (*p* < 0.001, Table 1).

The thyroid cancer group had a higher proportion of individuals with breast cancer than the control I group (Figure 3).

The adjusted HR for breast cancer in the thyroid cancer group was 1.64 (95% CI = 1.13–2.39, *p* = 0.010). In the subgroup analysis, the adjusted HR for breast cancer was higher in the thyroid cancer group than in the control I group in the ≥ 60-year-old subgroup. The adjusted HR for breast cancer was not significantly different between the thyroid cancer with the RAI treatment group and the control I group. The adjusted HR for breast cancer was higher in the thyroid cancer without RAI treatment group than in the matched control I group (Table 2).

#### 3.1.2. Study II. Risk of Thyroid Cancer in the Breast Cancer Group

Age, income, and region of residence were exactly matched between the breast cancer and control II groups (*p* = 1.000). The rates of high CCI scores, overweight, obesity I, obesity II, nonsmoking status and past smoker status, and alcohol consumption < 1 time a week were higher in the breast cancer group than in the control II group (each *p* <0.05). The ratio of thyroid cancer was higher in the breast cancer group than in the control II group (*p* < 0.001, Table 1). 

The breast cancer group had a higher proportion of individuals with thyroid cancer than the control II group (Figure 4).

The adjusted HR for thyroid cancer in the breast cancer group was 1.91 (95% CI = 1.47–2.49, *p* < 0.001). In the age subgroup analysis, the adjusted HRs for thyroid cancer were higher in the breast cancer group than in the control II group in all the age groups (Table 3).

## 4. Discussion

In our study, the HR for breast cancer in thyroid cancer participants was significantly higher than that in control I participants. This was consistent in the ≥60-year-old group and the group without RAI. Additionally, the HR for thyroid cancer in the breast cancer group was higher than that in control II participants and was constant in all subgroups. This study evaluated this reciprocal association using two different study procedures that analyzed data from a single cohort; we excluded participants with a previous history of breast cancer in the thyroid cancer group and with a previous history of thyroid cancer in the breast cancer group and applied an index date-matched study design. 

In our study, 1% of thyroid cancer patients (vs. 0.6% in the control I group) and 2.5% of breast cancer patients (vs. 1.4% in the control II group) developed second primary or concomitant breast cancer and thyroid cancer, respectively. An et al. analyzed 4243 thyroid cancer patients and 6833 breast cancer patients and compared age-matched control groups without second malignancies. Their results revealed that 4.3% of thyroid cancer patients and 2.6% of breast cancer patients were diagnosed with second primary or simultaneous breast cancer and thyroid cancer, respectively. They suggested that estrogen receptor (ER) or progesterone receptor (PR) signaling could be a common factor in the development of thyroid cancer and breast cancer [7]. 

However, Joseph et al. suggested that the increased risks of both breast cancer and thyroid cancer could be caused by treatment-related procedures or pathological processes associated with each cancer rather than common risk factors. The risk of developing thyroid cancer after breast cancer (17%) was much higher than the risk of developing breast cancer after thyroid cancer (2%). If these diseases were caused by common risk factors, the results of increased risk between thyroid cancer after breast cancer and breast cancer after thyroid cancer should have been similar. This suggests that there could be a more profound etiology, such as Cowden syndrome, which is known to increase the risks of both breast cancer and thyroid cancer [6,21]. 

Obesity is also associated with elevated risks of thyroid cancer and breast cancer and poor prognoses [14,15]. The underlying pathophysiology of the links connecting breast and thyroid cancer with obesity is not fully understood. Obesity causes insulin resistance, which is associated with prolonged hyperinsulinemia and increased circulating IGF1, which could exert neoplastic activity by promoting cell cycle progression and inhibiting apoptosis [22]. Moreover, insulin also directly influences tumor development through promotional and antiapoptotic effects. Smoking and drinking are common social habits and expose people to carcinogens known to cause several types of cancers. However, smoking and moderate drinking have been reported to have negative associations with the risk of thyroid cancer through mechanisms such as weight loss [23], reduced TSH levels [24], reduced thyroid volume [25], and altered sex hormone levels [26]. Smoking is negatively associated with breast cancer, except in those who smoke heavily prior to their first full-term pregnancy, due to DNA damage caused by carcinogens in cigarettes [13,27]. Consuming alcohol increases the risk of breast cancer by acting as a carcinogen and increasing estrogen levels [28]. Because obesity, smoking, and drinking could influence the association between breast and thyroid cancer, we performed adjustments for these common risk factors. However, there was still a reciprocal positive association between breast cancer and thyroid cancer.

There are several possible causes for the increased risk of breast cancer after thyroid cancer treatment. First, the accumulated dose of I-131 in radioiodine therapy could increase the incidence of second primary malignancies, such as breast cancer, leukemia, and other head and neck cancers. Fallahi et al. investigated 973 thyroid cancer patients treated with RAI. The incidence of second primary malignancy was significantly associated with the cumulative I-131 dose, especially when the dose exceeded 40 GBq [29]. Rubino et al. showed that the risk of a second primary malignancy, such as bone and soft tissue cancer, colorectal cancer, and salivary gland cancer, was increased in patients with a relatively high cumulative administered dose of I-131. However, the relative risk of breast cancer after I-131 treatment compared with no I-131 treatment was not elevated (RR 0.8; 95% CI, 0.5–1.1) [30]. Lang et al. reported that a cumulative radioiodine therapy dose of 3–8.9 GBq was the only independent risk factor for second primary cancer in thyroid cancer survivors [31]. However, Yu et al. analyzed 17 studies and found that prior RAI treatment did not increase the relative risk of breast cancer, salivary cancer, or hematologic malignancies [32]. In our study, prior RAI treatment did not significantly increase the HR for breast cancer. We think the impact of RAI may be small to lead to breast cancer, or the small number of patients who underwent RAI treatment resulted in insufficient statistical power to determine a significant difference in our study. Second, the stimulation of sodium–iodide symporter (NIS) expression in thyroid cancer patients before I-131 administration could increase NIS expression in the breast, resulting in an elevated risk of breast cancer. The stimulation of NIS expression before I-131 therapy could have a stronger influence than the radiation exposure dose on the increased risk of breast cancer in thyroid cancer [33]. Third, thyroid cancer patients have a long life expectancy with regular cancer screening. Therefore, breast cancer can be found incidentally. A previous history of cancer is related to an increased risk of secondary cancer development [34]. Fourth, an elevated thyroid hormone level (T3) is associated with the proliferation of breast cancer cells through the regulation of tumor suppressor proteins, P53 and pRb, as shown in an in vitro study [35]. However, the association between the level of thyroid hormone or thyroid-stimulating hormone and the risk of malignancy is still debatable.

Breast cancer patients had significantly elevated risks of a secondary malignancy, including ovarian cancer, thyroid cancer, and Non-Hodgkin lymphoma (NHL) in a Japanese study [36]. In Korea, after breast cancer treatment, the incidence rates of thyroid cancer, endometrial cancer, stomach cancer, bile duct cancer, and acute myeloid leukemia (AML) were significantly higher than in the general population [9,37]. Thyroid cancer is one of the representative malignancies that can be caused by radiation exposure. Grantzau, T. and Overgaard, J. reported that there was no significant association between radiotherapy for breast cancer and second primary thyroid cancer except in cases of childhood exposure [38]. Radiation therapy exposure did not increase the incidence of second primary thyroid cancer in patients over 30 years old [39]. 

This study has several advantages. First, in contrast to hospital-based studies, we included participants from a large representative nationwide population who had undergone health screening examinations. Moreover, this is the largest study to date to show the association between thyroid cancer and breast cancer over a relatively long follow-up period. Second, we calculated the HRs for thyroid cancer in the breast cancer group and breast cancer in the thyroid cancer group and compared them with those for well-matched control groups. The control groups were selected with a random number method and matched by age, income, and region of residence to prevent selection bias. Third, this is the first report based on the same database to evaluate the HR for thyroid cancer in the breast cancer group and breast cancer in the thyroid cancer group and their associations with age and RAI treatment after adjusting for possible risk factors such as the CCI score, cigarette smoking, alcohol consumption, and obesity.

This study has several limitations. First, detection bias could influence the elevated incidence of second primary cancer after both thyroid cancer and breast cancer. Patients with diagnosed cancer generally visit hospitals regularly, and the detection rate could be increased compared to that in the general population without cancer. Second, the number of RAI treatments was relatively small for achieving an appropriate statistical power. Moreover, due to the small number of RAI treatment participants, we could not perform subanalyses according to RAI dose. Third, we used patient claim codes from the Health Insurance Review and Assessment (HIRA) database to confirm which individuals were diagnosed with thyroid cancer with thyroidectomy and breast cancer. Fourth, we did not have data from patients who were younger than 40 years old because health screening examinations are performed on only individuals older than 40 years.

## 5. Conclusions

The adjusted HRs for breast cancer in the thyroid cancer group and thyroid cancer in the breast cancer group were significantly higher than those for control I and control II participants, respectively. The adjusted HRs for breast cancer in the thyroid cancer group were significantly increased in the subgroup over 60 years old and the subgroup without RAI treatment compared to those for the matched control I group.

## Figures and Tables

**Figure 1 jpm-12-00133-f001:**
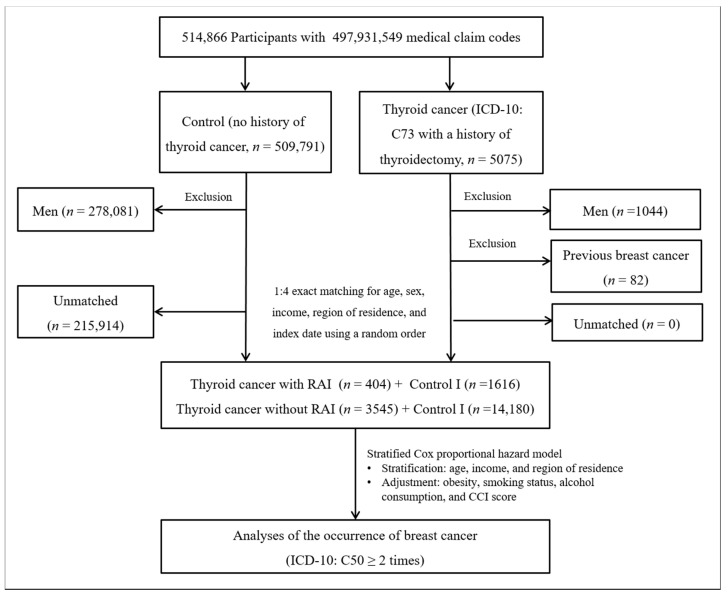
A schematic illustration of the participant selection process that was used in study I. Of a total of 514,866 participants, 3949 in the thyroid cancer group were matched with 15,796 in the control I group for age, income, and region of residence.

**Figure 2 jpm-12-00133-f002:**
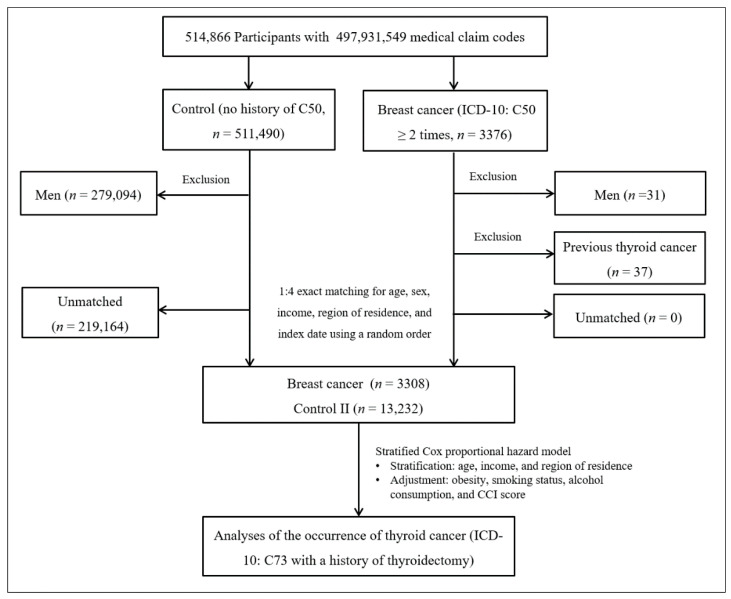
A schematic illustration of the participant selection process that was used in study II. Of a total of 514,866 participants, 3233 in the breast cancer group were matched with 12,932 in the control II group for age, income, and region of residence.

**Figure 3 jpm-12-00133-f003:**
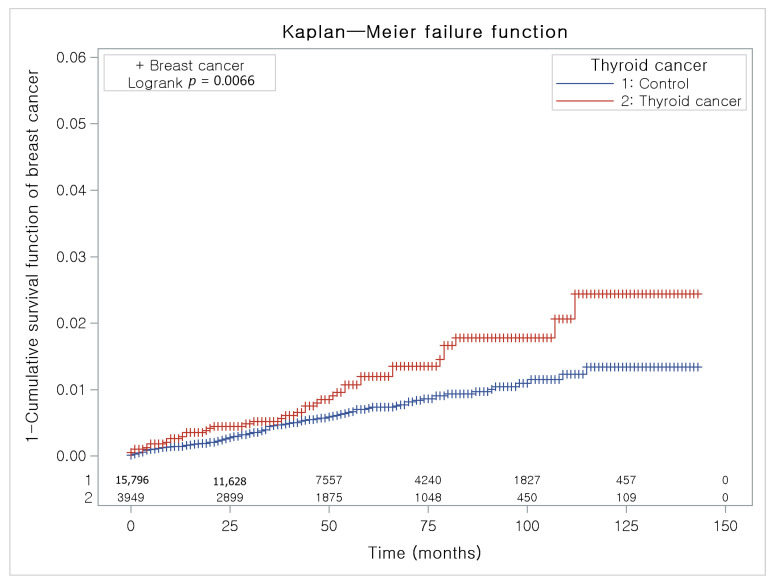
The Kaplan-Meier method was applied in study I. The rate of breast cancer was significantly higher in the thyroid cancer group than in the control I group.

**Figure 4 jpm-12-00133-f004:**
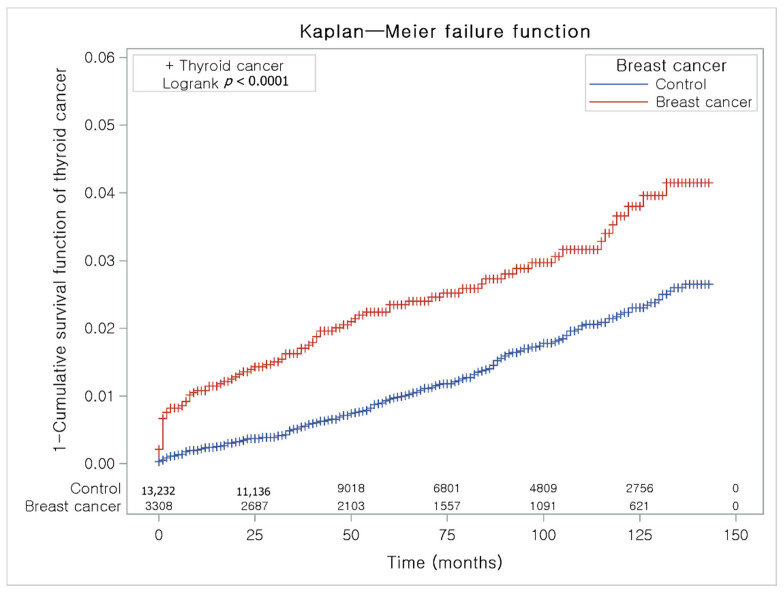
The Kaplan-Meier method was applied in study II. The rate of thyroid cancer was significantly higher in the breast cancer group than in the control II group.

**Table 1 jpm-12-00133-t001:** General characteristics of the participants.

Characteristics		Study I			Study II		
		Thyroid Cancer (*n*, %)	Control I (*n*, %)	*p*-Value	Breast Cancer (*n*, %)	Control II (*n*, %)	*p*-Value
Age (years old)				1.000			1.000
	40–59	2714 (68.7)	10,856 (68.7)		2307 (69.7)	9228 (69.7)	
	≥60	1235 (31.3)	4940 (31.3)		1001 (30.3)	4004 (30.3)	
Income				1.000			1.000
	Low (1–3)	1666 (42.2)	6664 (42.2)		1482 (44.8)	5928 (44.8)	
	High (4–5)	2283 (57.8)	9132 (57.8)		1826 (55.2)	7304 (55.2)	
Region of residence				1.000			1.000
	Urban	1919 (48.6)	7676 (48.6)		1651 (49.9)	6604 (49.9)	
	Rural	2030 (51.4)	8120 (51.4)		1657 (50.1)	6628 (50.1)	
CCI score				<0.001 *			<0.001 *
	0	3861 (97.8)	15,588 (98.7)		3165 (95.7)	13,034 (98.5)	
	1	9 (0.2)	37 (0.2)		17 (0.5)	47 (0.4)	
	≥ 2	79 (2.0)	171 (1.1)		126 (3.8)	151 (1.1)	
Obesity †				<0.001 *			0.018 *
	Underweight	53 (1.3)	333 (2.1)		275 (1.6)	54 (2.1)	
	Normal	1425 (36.1)	6236 (39.5)		5122 (36.5)	1208 (38.7)	
	Overweight	1100 (27.9)	4090 (25.9)		3475 (26.7)	882 (26.3)	
	Obese	1371 (34.7)	5137 (32.5)		1164 (35.2)	4360 (33.0)	
Current smoker		40 (1.0)	299 (1.9)	<0.001 *	55 (1.7)	297 (2.2)	0.015 *
Alcohol ≥ 1 time a week		204 (5.2)	1022 (6.5)	0.002 *	189 (5.7)	906 (6.9)	0.019 *
Breast cancer		39 (1.0)	94 (0.6)	0.007 *	3308 (100.0)	0 (0.0)	<0.001 *
Thyroid cancer		3949 (100.0)	0 (0.0)	<0.001 *	84 (2.5)	181 (1.4)	<0.001 *

*—Chi-square test. Significance at *p* < 0.05. † Obesity (BMI, body mass index, kg/m^2^) was categorized as <18.5 (underweight), ≥18.5 to <23 (normal), ≥23 to <25 (overweight) and ≥25 (obese).

**Table 2 jpm-12-00133-t002:** Crude and adjusted hazard ratios (95% confidence interval) for breast cancer in the thyroid cancer and control I groups.

Characteristics		Breast Cancer	Hazard Ratios for Breast Cancer			
		(Exposure/Total, %)	Crude †	*p*-Value	Adjusted †‡	*p*-Value
Total participants (*n* = 19,745)						
	Thyroid cancer	39/3949 (1.0)	1.67 (1.15–2.43)	0.007 *	1.64 (1.13–2.39)	0.010 *
	Control I	94/15,796 (0.6)	1.00		1.00	
Age < 60 years old (*n* = 13,570)						
	Thyroid cancer	27/2714 (1.0)	1.45 (0.93–2.25)	0.097	1.41 (0.91–2.20)	0.127
	Control I	75/10,856 (0.7)	1.00		1.00	
Age ≥ 60 years old (*n* = 6175)						
	Thyroid cancer	12/1235 (1.0)	2.55 (1.24–5.26)	0.011 *	2.56 (1.24–5.29)	0.011 *
	Control I	19/4940 (0.4)	1.00		1.00	
Thyroid cancer with RAI and controls (*n* = 2020)						
	Thyroid cancer	8/404 (2.0)	1.90 (0.82–4.41)	0.134	1.84 (0.78–4.30)	0.162
	Control I	17/1616 (1.1)	1.00		1.00	
Thyroid cancer without RAI and controls (*n* = 17,725)						
	Thyroid cancer	31/3545 (0.9)	1.62 (1.06–2.45)	0.024 *	1.59 (1.04–2.42)	0.031 *
	Control I	77/14,180 (0.5)	1.00		1.00	

*—Cox-proportional hazard regression model, Significance at *p* < 0.05; †—Models stratified by age, income, and region of residence. ‡—A model adjusted for obesity, smoking, alcohol consumption, and CCI scores.

**Table 3 jpm-12-00133-t003:** Crude and adjusted hazard ratios (95% confidence interval) for thyroid cancer in the breast cancer and control II groups.

Characteristics		Thyroid Cancer	Hazard Ratios for Thyroid Cancer			
		(Exposure/Total, %)	Crude †	*p*-Value	Adjusted †‡	*p*-Value
Total participants (*n* = 16,540)						
	Breast cancer	84/3308 (2.5)	1.96 (1.51–2.54)	<0.001 *	1.91 (1.47–2.49)	<0.001 *
	Control II	181/13,232 (1.4)	1.00		1.00	
Age < 60 years old (*n* = 11,535)						
	Breast cancer	63/2307 (2.7)	1.78 (1.32–2.38)	<0.001 *	1.72 (1.27–2.31)	<0.001 *
	Control II	150/9228 (1.6)	1.00		1.00	
Age ≥ 60 years old (*n* = 5005)						
	Breast cancer	21/1001 (2.1)	2.82 (1.62–4.92)	<0.001 *	2.91 (1.67–5.09)	<0.001 *
	Control II	31/4004 (0.8)	1.00		1.00	

*—Cox-proportional hazard regression model, Significance at *p* < 0.05; †—Models stratified by age, sex, income, and region of residence. ‡—A Model adjusted for obesity, smoking, alcohol consumption, and CCI scores.

## Data Availability

Release of the data by the authors is not legally allowed. Data in this study are available on the database of NHIS-HEALS in https://www.nhis.or.kr/ (assessed on 12 December 2020). NHIS permits access to all of these data via download for any researcher who promises to follow the research ethics.

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
