# Peer review of "Association between Thyroid Cancer and Breast Cancer: Two Longitudinal Follow-Up Studies Using a National Health Screening Cohort"

_jpm, 2022, doi:10.3390/jpm12020133_

Round 1
Reviewer 1 Report
The manuscript by Jin et al., is a retrospective study about the association between breast and thyroid cancers. The study was conducted in Korea collecting data obtained from 2002 to 2013, and the result of this analysis highlights a reciprocal positive correlation between these two cancers in females.
Overall, the content of this manuscript is of major interest. In my opinion, the manuscript is written very well, and I do not find any significant incorrectness. My following comments are of minor character.
- Line 30 and throughout the text: Please put the reference(s) before the point. For instance here should be …..the US [1]., instead of the US.[1]
- Lines 30-32: Please rephrase this sentence avoiding repetitions. Basically is identical to the previous sentence.
- Line 33: Please delete “female”
- Lines 36-38: Please rephrase this sentence avoiding repetitions. Basically is identical to the previous sentence.
- Lines 44-46: Please rephrase this sentence avoiding repetitions.
- Line 65: You should define the concept of “The hazard ratios (HR)”
- Please check the Reference list. Some references should be fixed
Reviewer 2 Report
The authors found that breast and thyroid cancer have a reciprocal positive association. It is very eye-catching to see that the authors identified the HR of breast cancer in thyroid cancer patients and then the HR of thyroid cancer in breast cancer.
